# Quantum Computing Approaches for Mission Covering Optimization

**Massimiliano Cutugno** [1,*], **Annarita Giani** [2], **Paul M. Alsing** [1] , **Laura Wessing** [1] **and Austar Schnore** [2]

1 Air Force Research Lab, Information Directorate, Rome, NY 13441, USA; paul.alsing@us.af.mil (P.M.A.); laura.wessing.1@us.af.mil (L.W.)
2 GE Research, General Electric, Niskayuna, NY 12309, USA
* Correspondence: massimiliano.cutugno@us.af.mil

**Abstract:** Quantum computing has the potential to revolutionize the way hard computational problems are solved in terms of speed and accuracy. Quantum hardware is an active area of research and different hardware platforms are being developed. Quantum algorithms target each hardware implementation and bring advantages to specific applications. The focus of this paper is to compare how well quantum annealing techniques and the QAOA models constrained optimization problems. As a use case, a constrained optimization problem called mission covering optimization is used. Quantum annealing is implemented in adiabatic hardware such as D-Wave, and QAOA is implemented in gate-based hardware such as IBM. This effort provides results in terms of cost, timing, constraints held, and qubits used.

**Keywords:** quantum computing; quantum annealing; NISQ devices; constrained optimization; constraint satisfaction problems

## 1. Introduction

Quantum computing hardware is quickly developing [1] together with the theory of quantum computing algorithms [2]. Problems historically solved classically have been translated into quantum computing formulations [3–6], and these formulations differ even within the various quantum hardware types. In this work, we explore how well constrained optimization can be implemented between two well-known quantum computing hardware types, (i) adiabatic, and (ii) the gate-based. Many constrained optimization problems can be formulated into Quadratic Unconstrained Binary Optimization (QUBO) in which adiabatic and gate-based hardware types solve via quantum annealing and QAOA, respectively.

To perform our analysis, we choose a constrained optimization problem that can be implemented on current gate-based and annealing hardware. Job-shop scheduling [7] is one particular NP-Hard constrained optimization problem with many applications in industry; it is intractable to obtain exact solutions on classical hardware. There are efforts to compose job-shop scheduling onto quantum hardware [8]; however, this requires many ancillary qubits to encode the many constraints that come with the problem. This is not ideal for current gate-based hardware, as the machines that are used are very limited in qubits. Another industry-useful NP-Hard problem is the generalized assignment problem [9,10]. The formulation of general assignment on gate-based architecture is also bottlenecked by the qubit amount due to the inequality constraints that are encoded within it. Motivated by the applicability of these classically intractable problems, we construct a new optimization problem we coin as Mission Covering Optimization (MCO) that we believe meets the criteria for our analysis as it is implementable on current quantum hardware (both adiabatic and gate-based) while still having applicability (when scaled) in the industry. Furthermore, specific scenarios of MCO can be designed to set the number of constraints. This is important for analyzing how the implementation may change when additional constraints are added to each problem.

The goal of this paper is to assess the performance of each algorithm used when solving a particular Mission Covering Optimization problem. Variations of MCO problems are of great interest to industry as well as government. Manufacturing operations, for example, require covering a set of steps with a number of needed resources. Airline network optimization consists in best allocating airplanes, pilots, flight attendants, and maintenance crew [11]. Military missions success is the result of a large number of resources working together for a common goal. Optimizing these processes will bring great societal benefits.

MCO describes a problem in which a set of missions are to be performed given constraints on the available resources, where missions are organized efforts to achieve an objective. Examples of mission types include industrial, logistic, and militaristic, where the problem can be anything from transporting packages to destinations to efficiently producing and maintaining equipment devices [8,12]. The global solutions to such problem are computationally challenging. It is a complex optimization problem [13], often non-convex. Heuristic approaches are used to find feasible solutions, but they depend on domain knowledge and problem decomposition, leading to inaccurate solutions. Therefore, this category of problems is a good candidate for quantum acceleration. In this effort, we will address an abstraction of a real MCO problem.

We formulate the problem for a Quantum Annealer, specifically the D-Wave quantum computer, as well as investigate the formulation for a gate-based model approach, specifically the IBM quantum computer. By using the results from the Quantum Processing Unit (QPU) and through simulations, we present strengths and weaknesses found with both models, and outline the fundamentals of this new type of optimization problem.

### 1.1. Quantum Solutions to MCO

As briefly introduced before, we use three types of optimization problems in this study on the two different types of quantum machines. We summarize each technique in the following text.

#### 1.1.1. Quantum Annealing (QA)

Quantum Annealing is a quantum optimization methodology performed on a specific implementation of quantum computers such as D-Wave quantum annealing machines. It is an adiabatic quantum computing technique that capitalizes on a unique feature of quantum mechanics, e.g., quantum tunneling, which is the capability to surf and dive through an energy landscape until it hits a minimum energy level [14]. This property of quantum mechanics is engineered to solve Quadratic Unconstrained Binary Optimization problems [15]. QUBO is a type of optimization problem where the solution is a binary vector that minimizes an objective function described with terms up to a quadratic degree. These problems are unconstrained, but there are methods to incorporate the behavior of constraints [16] into the cost function. To encode an MCO onto a quantum annealing machine, we must translate the objective function into a QUBO.

#### 1.1.2. Quantum Alternating Operator Ansatz (QAOA)

Quantum Alternating Operator Ansatz (QAOA) is a method for solving combinatorial optimization problems on NISQ devices [17]. The algorithm is supported by gate-based quantum computers such as the IBM, Rigetti, IonQ or Xanadu machines. In this effort, we focus on implementing the problem on an IBM quantum computer [18]. At the current time, IBM's Qiskit implementation of QAOA uses a Variational Quantum Eigensolver (VQE) to find the expectance of a parameterized ansatz eigenstate. This quantity is then used to calculate the minimum of the cost function, which is embedded in a cost Hamiltonian $H_C$. Similar to QA, it is limited to solving binary quadratic problems [19].

#### 1.1.3. Quantum Alternating Operator Ansatz with Constraint Hamiltonian (QAOAH)

In QAOA, the starting state is in an equal superposition of all computational basis states, and the mixing Hamiltonian is a sum of Pauli-X operators acting on each qubit.

In this configuration, every possible classical-state solution can be traversed, making it ideal for unconstrained problems where there are no states to be filtered out. However, it is possible to incorporate constraints by altering the default mixing Hamiltonian $H_M$ and the initial quantum state $|\psi\rangle$ [20]. Since the mixing operator (the exponentiated mixing Hamiltonian) specifies how to explore the solution space, it is possible to specify a mixing operator that determines how to move to another solution within the constraints of the problem.

Consider an initial quantum state $|\psi\rangle$, which represents a solution that does not violate the constraints of the optimization problem but is not necessarily the minimizer. Consider the mixing operator $H_M$ that is constructed such that after an application on a state within the constrained space, the output state is also guaranteed to be within the constrained space. The idea is to construct $H_M$ and $|\psi\rangle$ so that when stopping QAOA at any iteration, the final encoded solution is always within the constrained solution space. If this mixing operator can mix such that it allows the algorithm to reach every possible classical-state in the constrained solution space, it is then possible to obtain the minimizer in that same space [20].

*1.2. Paper Outline*

This paper is divided into four main sections. Section 2 describes MCO in greater formal detail. Section 3 outlines two different scenarios of MCO that are tested in this study. Section 4 details how each scenario is implemented using three different algorithmic techniques. Section 5 describes the results of each scenario and compares the different algorithmic techniques by cost, timing, and constraint-holding metrics. In Section 6, we summarize and conclude our findings of this work. Lastly, Section 7 discusses research directions for future work.

## 2. Formalism for Mission Covering Optimization

The objective of an MCO problem is to allocate a set of resources to missions such that each resource is assigned to at most one mission, and each mission's requirements are satisfied.

An MCO problem is described by a 7-tuple:

$$(\mathbf{M}, \mathbf{R}, \mathbf{Q}, \mathsf{C}, \mathrm{REQ}_M, \mathrm{REQ}_R, f_{obj})$$

1.  $\mathbf{M}$ is the set of missions,
2.  $\mathbf{R}$ is the set of resources,
3.  $\mathbf{Q}$ is the set of qualifications,
4.  $\mathsf{C}$ is the capability function,
5.  $\mathrm{REQ}_M$ is the mission's requirements function,
6.  $\mathrm{REQ}_R$ is the resource's requirements function,
7.  $f_{obj}$ is the objective function that scores problem solutions.

In the rest of this section, these terms will be explained in detail.

*2.1. Missions*

Missions are operations to achieve specific results. They require resources for specific tasks. For example, a mission could be transporting children to school. The resources are the buses and the bus drivers. In manufacturing, a mission could represent the process of building an asset from design to delivery. It includes design, engineering, sourcing, suppliers, production, control, and packaging. $N_M$ denotes the number of missions. In an MCO, missions need to be completed at the same time; we denote the mission set as $\mathbf{M} = \{m_1, m_2, \ldots, m_{N_M}\}$.

### 2.2. Resources

**R** represents the set of all resources. Resources are all the assets needed for mission completion. Resources can be objects, machines or people. For example, a given MCO may define its resources as four planes, four pilots and two engineers. $\mathbf{R} = \{r_1, r_2, \ldots, r_{N_R}\}$ is the set of $N_R$ resources within the MCO.

### 2.3. Qualifications

**Q** represents the set of all qualifications. Resources are specialized in the sense that they have qualifications, and they can be assigned to the mission that requests a resource with such qualification. For example, 'Can transport', 'Can pilot', and 'Can troubleshoot' are qualifications a mission may require. $N_Q$ is the number of qualifications in the set $\mathbf{Q} = \{q_1, \ldots, q_{N_Q}\}$.

### 2.4. Capabilities

C represents the capability function. Each resource is scored for its qualifications. Capabilities are integer numbers that represent how well a resource can perform a certain qualification. The higher the number, the more qualified the resource is. For example, consider the resource set composed of a senior engineer, intern engineer, and an HR Manager and with capabilities of 2, 1, 0, respectively, to a certain qualification titled 'troubleshoot carburetor'. This reflects the fact that the senior engineer has a higher capability than the intern engineer to 'troubleshoot carburetor' because he/she is more experienced in the field. Conversely, the HR manager has no qualification capability to 'troubleshoot carburetor'; therefore, has a score of zero. Capability is the function

$$\mathsf{C}\colon \mathbf{R} \times \mathbf{Q} \to \mathbb{Z}_{\geq 0},$$

which returns zero if and only if the resource is not qualified for the specified qualification. When this happens, we say that this resource has no capability for that qualification.

Missions require resources with specific qualifications. For example, mission $m_1$ requires $n$ resources with qualification $q_1$ where $m_1 \in \mathbf{M}$, $n \in \mathbb{N}$ and $q_1 \in \mathbf{Q}$, which represents 'mission 1 requires $n$ resources that can fly a plane' ($q_1$). We assume that resources with higher capability for certain qualifications are chosen.

### 2.5. Mission's Requirements

The mission's requirement function is described as

$$\mathsf{REQ}_M\colon \mathbf{M} \times \mathbf{Q} \to \mathbb{N},$$

This function returns the number of required resources that satisfy the qualification needed for that mission.

### 2.6. Resource's Requirements

The resource's requirement function is described as

$$\mathsf{REQ}_R\colon \mathbf{R} \times \mathbf{Q} \to \mathbb{N},$$

This function returns the number of required resources that satisfy the qualification needed for that resource.

### 2.7. Solutions and Solution Score

A solution of the MCO is defined as a function that associates resources to missions

$$f_s\colon \mathbf{R} \to \mathbf{M},$$

If $N_F$ is the number of functions that are the solutions of the MCO, then $\mathbf{F} = \{f_1, f_2, \ldots, f_{N_F}\}$ is the set of all possible solutions. The goal is to find the best solution in which all missions meet requirements as well as possible. The objective function maps solutions to real numbers, which measures how far off each solution is from meeting requirements. The objective function represents a cost function:

$$f_{obj} \colon F_s \to \mathbb{R}$$

s.t.

$$F_s = \{\{(m, r) \in \mathbf{M} \times \mathbf{R} \mid m = f_s(r)\} \mid f_s \colon \mathbf{R} \to \mathbf{M}\}.$$

Therefore, the best solution to an MCO is achieved by minimizing the objective function and retrieving the minimizer.

## 3. MCO Scenarios

The MCO optimization problem involves the parameters (missions, resources, mission require, resource require). From this, there are different metrics of cost that can be used to represent the objective function. Two specific MCO scenarios are formalized and implemented on different quantum computing hardware machines. These scenarios are not computationally difficult to compute classically. However, the design of these scenarios is intentional for comparing results in this study, as brute force methods for finding the absolute best cost-minimizers can be used without being throttled by the exponential time complexity since the scenarios are designed to be permutationally symmetric. The primary focus of each of these readily solvable MCO scenarios is to identify how well each algorithm implementation performs when constraints are introduced. Results are compared in terms of cost, timing, and constraints met. In the general MCO formulation, resources can be assigned to at most one mission so that resources could be unused. In this paper, in both scenarios, it is assumed that each resource is assigned to one and only one mission. This is acheieved by introducing an additional mission that includes the unused resources. The following subsections describe each scenario.

### 3.1. Scenario 1: Primary and Secondary Resources

The first scenario involves two categories of resources: primary and secondary. Primary resources are ready to be used for mission completion. Secondary resources are allocated when primary resources are not able to perform their duty. Primary resources that cannot be allocated to missions are removed from the set of primary resources. For example, the mission covering optimization solution shown in Figure 1 is composed of three missions requiring three, two and two resources, respectively, for mission completions. Real-world applications are aeronautical missions and operations with pilots as resources. This particular example is used throughout the paper. Suppose in the planning phase that seven pilots were allocated to cover the three missions; they are on-duty (in the primary resource set), and three other pilots are on-call (in the secondary resource set). Suppose at a certain point in time, two of the pilots became sick, and they were removed from the resource group. This is an emergency, an unforeseen situation that requires re-running an optimization algorithm to cover the missions, including secondary resources. The pilots are pulled from the set of five pilots on-duty, ready to cover the missions and two pilots from the secondary resource set should also be included. The challenge is to find the best allocation of pilots to missions using all the pilots on-duty and two pilots on-call. The goal of our effort is to use quantum computing to solve this problem and analyze results from different quantum computing hardware implementations.

The following outlines all the rules in this specific scenario:

1. The set of missions in the problem is $\mathbf{M} = \{m_1, \ldots, m_{N_m-1}, U\}$.
2. The set of resources is $\mathbf{R} = \{r_1, \ldots, r_{N_r}\}$.
3. There only exists one qualification, which is the ability to fly a plane, $\mathbf{Q} = \{q_1\}$.

4. Resources can have a capability of 1 or 2. Therefore, the capability function is then $C: \mathbf{R} \times \mathbf{Q} \rightarrow \{1, 2\}$. This is a way to represent how ready the resource is to be allocated to a mission. It can be thought of as an ordering for the allocation of resources; resources with capability 2 should be allocated before ones with 1. Resources that have capability 2 are referred to as primary resources, which are assigned to missions first. Resources that have capability 1 are referred to as secondary resources, which are assigned to missions only if primary resources become unavailable.

5. $REQ_M$ is the mission's requirement. For example:

   - $REQ_M(m_1, q_1) = 3$.
   - $REQ_M(m_2, q_1) = 2$.
   - $REQ_M(m_3, q_1) = 2$.

6. There are no resource requirements. Therefore, the resource requirement function is

$$REQ_R: \mathbf{R} \times \mathbf{Q} \rightarrow \{0\}.$$

7. $f_{obj}$ is the function used to score the different problem solutions in terms of cost The mission cost includes two parts:

   - The mission cost represents how many of the mission's requirements are not met. It is a penalty introduced every time a mission is not able to accomplish its goal due to a lack of resources.
   - The precedence cost measures how well each solution allocates resources with higher capabilities before others to missions. In the specific example, it means that it is desirable that primary pilots are allocated to missions before secondary pilots.

As discussed previously, MCO covers a broad spectrum of problems distinguishable only by its objective function. In the scenario described here, the objective function reflects the cost, which is the sum of the mission cost and precedence cost. All solutions are measured in terms of the cost in order to find the optimal solution. This scenario can be described by using a matrix arrangement of Boolean variables, shown in Figure 1. Each row of the table in the figure represents a mission, and each column represents a resource.

**Figure 1.** Matrix view for Solution: This represents a table representing the solution of a problem with three missions, five primary resources and three secondary resources. A purple circle symbolizes a Boolean variable $x_{m,r} \in \{0, 1\}$ where $m$ represents the mission (row) while $r$ represents the resource (column). A circle filled in with orange represents $x_{m,r} = 1$ and the fact that resource $r$ is allocated to mission $m$.

The final row of the table represents an artificial mission created for the purpose of having all the resources utilized. When resources are not allocated to any of the other missions, it has to be allocated this special mission. It will be referred to as the unallocated mission (U mission for short) and has no mission cost associated.

A separate column specifies the required amount of qualifying resources for each mission. Since there is only one qualification in the scenario, this number just represents

how many resources should be allocated to the mission. The example in Figure 1 shows a solution where each mission satisfies its requirements, but only after using all primary resources before it uses any of the secondary ones. Since each resource can only be assigned to one mission, no more than one Boolean variable may be true in a single column of the matrix representation of an MCO. In general, the column constraint assures that a resource can be assigned at most to one mission. This is a hard constraint.

### 3.2. Scenario 2: The Buddy System

This scenario was designed to see how different algorithms deal with additional constraints. There are two groups of resources of different types. If a resource from one group is assigned to a mission, it is required that a resource from the other group joins the same mission. Consider the previous example where a set of planes and pilots must be allocated to a set of aerial missions. A pilot and plane are required to complete a mission. Every time a plane is chosen, a pilot needs to be chosen. From an application standpoint, this scenario highlights the modeling of resource dependencies (planes depend on pilots for allocation). This scenario contains an additional row constraint. The list of constraints is as follows:

- Column constraint: a resource can be assigned at most to one mission. This is the hard constraint of the problem.
- Row constraint: if a type of resource is chosen, then a resource of the second type must also be chosen. We call it the *buddy* constraint. This is the additional constraint added in this scenario.

The entire set of rules for this scenario is as follows:

1. The set of missions in the problem is $\mathbf{M} = \{m_1, \ldots, m_{N_m-1}, U\}$.
2. The set of resources is $\mathbf{R} = \{r_1, \ldots, r_{N_r}\}$.
3. There are two qualifications $\mathbf{Q} = \{q_1, q_2\}$, which means that resources are divided in two groups: $\mathbf{R}_1$ and $\mathbf{R}_2$.
4. Resource's capability is $C \colon \mathbf{R} \times \mathbf{Q} \to \{0, 1\}$, which means that all resources that are qualified have the same capability. Resources have only one qualification:
   - The set of all resources that have capability 1 for qualification $q_1$ is notated as $\mathbf{R}_1$.
   - The set of all resources that have capability 1 for qualification $q_2$ is notated as $\mathbf{R}_2$.
   - The sets $\mathbf{R}_1$ and $\mathbf{R}_2$ do not contain the same resource: $\mathbf{R}_1 \cap \mathbf{R}_2 = \{\}$.
   - The sets $\mathbf{R}_1$ and $\mathbf{R}_2$ contain all resources: $\mathbf{R}_1 \cup \mathbf{R}_2 = \mathbf{R}$.
   - The number of resources in $\mathbf{R}_1$ and $\mathbf{R}_2$ is the same $|\mathbf{R}_1| = |\mathbf{R}_2|$.
5. Missions require resources with qualification $q_1$ and not $q_2$. Therefore, the mission require function can be formally described as: $\mathrm{REQ}_M \colon \mathbf{M} \times \{q_1\} \to \mathbb{Z}_{\geq 0}$.
6. The resource's requirement is called the *buddy requirement*. Every resource when allocated requires another resource with the opposite qualification.
   - $\mathrm{REQ}_R(r, 1) = 0$ & $\mathrm{REQ}_R(r, 2) = 1 \quad \forall r \in \mathbf{R}_1$
   - $\mathrm{REQ}_R(r, 1) = 1$ & $\mathrm{REQ}_R(r, 2) = 0 \quad \forall r \in \mathbf{R}_2$
7. As in the previous scenario, the objective function measures the total mission cost.

The mission cost formulation is exactly the same as the secondary resources scenario. However, the way we view the problem in its matrix representation is slightly different, as shown in Figure 2.

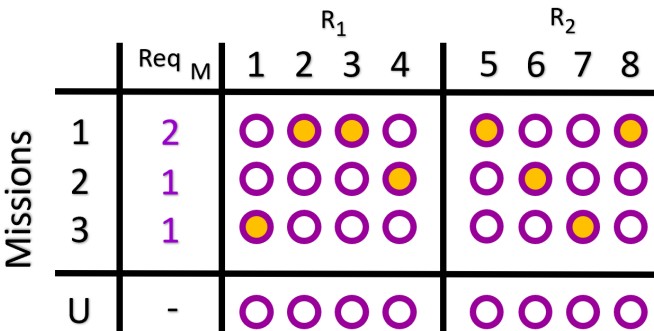

**Figure 2.** Matrix view for Solution. This solution is shown for a problem with three missions, four resources of type $\mathbf{R}_1$ and four resources of type $\mathbf{R}_2$. REQ$_M$ describes how many $\mathbf{R}_1$ resources must be assigned to mission $m$. Each $\mathbf{R}_1$ resource is paired with exactly one $\mathbf{R}_2$ resource for each mission.

The primary difference between this solution and the prior scenario's solution is that now there are *buddy* constraints present along the rows of the matrix (for this reason, it is also called the *row* constraint). For example, in Figure 2, the first row related to the first mission has two resources allocated from $\mathbf{R}_1$. In order for the buddy constraint to not be violated, two resources from $\mathbf{R}_2$ are allocated to that mission. The solution shown is valid since no rows or columns violate any of the constraints. The solution also has the lowest mission cost since the mission requirements were all exactly met.

Finding optimal solutions for this scenario is trivial since all resources within the same group are indistinguishable from each other in terms of allocation cost. However, when adding different capabilities to resources within this MCO, the complexity of finding the optimal solution increases. The case where resources have different and multiple capabilities has not been studied and is part of future work. While this does not substantially change how the problem is formulated and implemented on the quantum device, it is a more complex method of checking how well the quantum algorithms solutions are performing compared to the optimal solution. To do this, the optimal solution needs to be known, so brute force methods are used to find it. If resources are indistinguishable from one another, then it takes far less time to brute force the optimal solution due to the permutation symmetry across resources.

## 4. Algorithm Implementation

In this work, we tested three techniques to solve the MCO problem with the two scenarios described in the previous section, quantum annealing and two types of QAOA algorithms.

- Quantum Annealing (QA) [21]
- Quantum Alternating Operator Ansatz (QAOA) [17]. It implements constraints by means of a Lagrange multiplier embedded into a cost function Hamiltonian.
- QAOAH, which is a version of QAOA with a constrained mixer. It is denoted as QAOAH as it was developed by Hadfield et al. [20]. It engineers the constraints to remain within a constraint space during the entire solution process.

All three approaches use similar means for encoding the objective function but differ in the way they implement their constraints and what machines support them.

Since QA and QAOA are both derived from the same formulation, the implementation details are described together. In QAOAH, constraints are embedded into the Mixing Hamiltonian, which results in a different formulation. The following sub-sections outline the implementation for both scenarios using the three approaches described above.

*4.1. Scenario 1 (One Constraint)*

4.1.1. QA and QAOA

The goal of the MCO problem is to minimize the total cost. Various methods are used to define the mission cost. The following formulation takes into account the limitations of QUBO. In the rest of the paper, $QA$ denotes QUBO problems performed on quantum annealing machines, while QAOA and QAOAH denote QUBO problems executed on gate-based architectures.

The optimal mission cost occurs when all the resources required are allocated. Or, formally, consider mission $m$; the optimal mission cost occurs when:

$$\sum_{r \in \mathbf{R}} x_{m,r} = \mathsf{REQ}_M(m, 1). \tag{1}$$

The $x$ values represent Booleans (an alternative representation of the solution $f_s$), such that when indexed by $m \in \mathbf{M}$ and $r \in \mathbf{R}$, e.g., $x_{m,r}$, it represents whether or not the solution mapped resource $r$ to mission $m$.

As an example, suppose there are three resources, $\mathbf{R} = \{1, 2, 3\}$ and the first mission requires two of them, then the optimal mission cost for the mission occurs when:

$$x_{1,1} + x_{1,2} + x_{1,3} = 2. \tag{2}$$

Equation (2) is true when two out of the three variables are true. The mission cost represents the penalty added any time a mission lacks one or more needed resources. The penalty is higher when more resources are missing. A squared error is used to represent the mission cost:

$$\mathsf{MC}(x, m) = \left( \sum_{r \in \mathbf{R}} x_{m,r} - \mathsf{REQ}_M(m, 1) \right)^2, \tag{3}$$

where $\mathsf{MC}(x, m)$ is the cost associated with mission $m$. It can be seen from Equation (3) that the minimum mission cost, when $\mathsf{MC}(x, m) = 0$, yields the optimal case, as described in Equation (1). The mission cost is quadratic. The example that yielded the optimal case in Equation (2) in terms of the mission cost function is:

$$\mathsf{MC}(x, 1) = (x_{1,1} + x_{1,2} + x_{1,3} - 2)^2. \tag{4}$$

For this optimization problem, secondary resources should be allocated only after primary resources are allocated first. Therefore, we introduce precedence costs for resources. Equation (5) shows the ideal condition regarding the allocation of secondary resources.

$$\sum_{\substack{m \in \mathbf{M} \\ m \neq U}} x_{m,r} = \mathsf{C}(r, 1) - 1 \quad \forall r \in \mathbf{R} \tag{5}$$

The precedence cost is dependent on the capability of a resource and, therefore, dependent on whether or not this resource is primary or secondary. When the resource is primary, the ideal condition means that the it must be allocated to one of the active missions (apart from the unallocated/psuedo mission). For the other secondary resources, ideally none are used.

The precedence cost may or may not be met when optimizing, so similarly to mission cost, precedence cost is the squared error of this equality:

$$\mathsf{PC}(x, r) = \left( \sum_{\substack{m \in \mathbf{M} \\ m \neq U}} x_{m,r} - \mathsf{C}(r, 1) + 1 \right)^2. \tag{6}$$

Finally, the objective function is then the total mission cost and the total precedence cost in the MCO:

$$f_{obj}(x) = \sum_{m \in \mathbf{M}} \mathsf{MC}(x, m) + \frac{1}{|\mathbf{R}|} \sum_{r \in \mathbf{R}} \mathsf{PC}(x, r). \tag{7}$$

We weight the precedence cost by $\frac{1}{|\mathbf{R}|}$ to ensure that the mission cost is minimized before precedence cost. The total cost is the sum of the mission cost and the precedence cost, and it is reflected in the objective function.

The constraint that a resource must be paired to exactly one mission is formulated:

$$\sum_{m\in\mathbf{M}} x_{m,r} = 1 \quad \forall r \in \mathbf{R}. \tag{8}$$

The constraint function used for this scenario is defined as

$$\mathsf{CONSTR}(x,r) = \left(\sum_{m\in\mathbf{M}} x_{m,r} - 1\right)^2. \tag{9}$$

Since the QUBO is used to solve problems without constraints, we must add it to the objective function so that when it minimizes, the constraints will be met. The method of Lagrange multipliers is a strategy for finding the local maxima and minima of functions subject to equality constraints.

If $f_{obj}(x)$ is the objective function to be minimized, the Lagrangian function is

$$\mathcal{L}(x,\lambda) = f_{obj}(x) + \lambda \cdot \mathsf{CONSTR}(x). \tag{10}$$

The solution to the original constrained problem is always a saddle point of this function. Setting a large value for $\lambda$, the term related to the constraint, will have the greatest impact on the optimization problem. The solution will minimize the constraint first and then the cost.

The new objective function that includes the constraints is:

$$\begin{aligned} f_{obj}(x) &= \mathcal{L}(x,\lambda) \\ &= \sum_{m\in\mathbf{M}} \mathsf{MC}(x,m) + \frac{1}{|\mathbf{R}|}\sum_{r\in\mathbf{R}} \mathsf{PC}(x,r) \\ &\quad + \lambda \cdot \sum_{r\in\mathbf{R}} \mathsf{CONSTR}(x,r). \end{aligned} \tag{11}$$

QAOA for this method uses an equal superposition for the starting state $|\Psi\rangle$ over $N$ states:

$$|\Psi\rangle = \frac{1}{\sqrt{N}}\sum_{i=0}^{N-1}|i\rangle. \tag{12}$$

The value $N$ is equal to $2^n$ where $n$ is the number of qubits used. For this problem, the number of qubits used is $N_M \cdot M_R$, and the number of missions times the number of resources.

The mixing operator is constructed using a Hamiltonian, which is the sum of Pauli-X as follows:

$$X_i = \underset{1}{I} \otimes \cdots \otimes \underset{i}{X} \otimes \cdots \otimes \underset{n}{I}, \tag{13}$$

$$H_M = \sum_{i=1}^{n} X_i. \tag{14}$$

### 4.1.2. QAOAH

In the last section, Lagrange multipliers are used to encode constraints into the QUBO problem. Alternatively, by choosing the appropriate mixing Hamiltonian and initial state, we can constrain the solution space outside of the QUBO formulation in QAOA [20].

The initial state must be within the constrained solution space. A trivial starting configuration that is known not to violate the constraint is when all resources are set to the unallocated mission, as shown in Figure 3.

## Resources

| | Req $_M$ | Primary | | | | | Secondary | | |
|---|---|---|---|---|---|---|---|---|---|
| Missions | | 1 | 2 | 3 | 4 | 5 | 6 | 7 | 8 |
| 1 | 2 | ◯ | ◯ | ◯ | ◯ | ◯ | ◯ | ◯ | ◯ |
| 2 | 1 | ◯ | ◯ | ◯ | ◯ | ◯ | ◯ | ◯ | ◯ |
| 3 | 1 | ◯ | ◯ | ◯ | ◯ | ◯ | ◯ | ◯ | ◯ |
| U | - | ● | ● | ● | ● | ● | ● | ● | ● |

**Figure 3.** Example Initial State: All resources are initialized to be unallocated, or allocate to mission $U$, the last row.

This initial state used is:

$$|\psi\rangle = \bigotimes_{(m,r)\in \mathbf{M}\times\mathbf{R}} \begin{cases} |1\rangle & m = U \\ |0\rangle & m \neq U \end{cases}. \tag{15}$$

The mixing Hamiltonian describes how to move from the starting state, as well as all subsequent states, such that they are also in the constrained space. To be in the constrained space, only one qubit per column must be active. The Hamiltonian should describe how to cycle a qubit in an active state throughout the column so that it can reach every possible combination of configurations that still satisfy the constraint. The identity operator and the SWAP gates are used for this cycling action. For the thre mission examples, we can confirm that an individual column can have each of its possible states reached using three swap gates (see Figure 4).

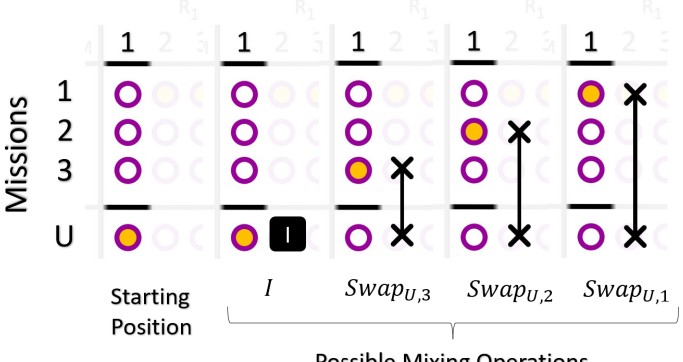

**Figure 4.** Resource 1's Mixing Operators: The leftmost segment details the starting state before applying the mixing operator. The second segment from the left shows an identity mixing operator that allows the current state to remain unchanged. The three right-most segments show an operation that uses one swap to cycle the qubit's on state to missions 1, 2, and 3.

Thus, a single resource mixing MIX($r$) is:

$$\text{MIX}(r) = \sum_{\substack{m\in\mathbf{M} \\ m\neq U}} \text{SWAP}_{(U,r),(m,r)}, \tag{16}$$

where $(m,r)$ encodes the index of the qubit representing a mapping of resource $r$ and mission $m$. The $U$ in $(U,r)$ represents the unallocated mission. The total mixing operator $H_m$ is the sum of MIX($r$) on each resource with the identity operation:

$$H_m = I^{\otimes n} + \sum_{r \in \mathbf{R}} \mathsf{MIX}(r). \tag{17}$$

In order to embed the mixing operation as a Hamiltonian to run on the IBM machines, it must be described as a composition of tensored Pauli gates. Each SWAP gate can be decomposed in terms of Pauli gates:

$$\mathsf{SWAP} = \frac{1}{2}(I \otimes I + X \otimes X + Y \otimes Y + Z \otimes Z). \tag{18}$$

When SWAP is indexed by $i$ and $j$, the Pauli gates fall on the $i$th and $j$th, qubits respectively:

$$\mathsf{SWAP}_{i,j} = \frac{1}{2}\left(I^{\otimes n} + X_{i,j} + Y_{i,j} + Z_{i,j}\right), \tag{19}$$

$$X_{i,j} = \underset{1}{I} \otimes \cdots \otimes \underset{i}{X} \otimes \cdots \otimes \underset{j}{X} \otimes \cdots \otimes \underset{n}{I} \ . \tag{20}$$

Each Pauli gate used in Equation (19) ($X_{i,j}$, $Y_{i,j}$, $Z_{i,j}$) is defined similarly to what is defined in Equation (20). For clarity, Equation (20) places the corresponding Pauli gate only at the specified indices $i$ and $j$ in a tensor product of identities $I$.

### 4.2. Scenario 2 (Two Constraints)
#### 4.2.1. QA and QAOA

In this scenario, the objective function measures just the mission cost, as opposed to the previous scenario that also measures precedence cost. Therefore, the objective function is the sum of the mission cost and the constraint function:

$$f_{obj}(x) = \sum_{m \in \mathbf{M}} \mathsf{MC}(x, m) + \lambda \cdot \mathsf{CONSTR}(x). \tag{21}$$

Two different constraints are embedded in the objective function using Lagrange multipliers. The first one was discussed in the previous scenario and assures that resources are allocated to no more than one mission. It is formulated in a similar way as before:

$$\mathsf{CONSTR}_1(x, r) = \left(\sum_{m \in \mathbf{M}} x_{m,r} - 1\right)^2. \tag{22}$$

The additional constraint (buddy constraint) requires that the amount of resources allocated from set $\mathbf{R}_1$ must be the same as the number of resources allocated from set $\mathbf{R}_2$. This hard equality is formulated as:

$$\mathsf{CONSTR}_2(x, m) = \left(\sum_{r \in \mathbf{R}_1} x_{m,r} - \sum_{r \in \mathbf{R}_2} x_{m,r}\right)^2. \tag{23}$$

The total constraint function expressed in Equation (21) can be expressed as the sum of both of these constraints:

$$\begin{aligned} \mathsf{CONSTR}(x) = \ & \sum_{r \in \mathbf{R}} \mathsf{CONSTR}_1(x, r) \\ & + \sum_{\substack{m \in \mathbf{M} \\ m \neq U}} \mathsf{CONSTR}_2(x, m) \end{aligned} \ \ . \tag{24}$$

For QAOA, the starting state and the mixing Hamiltonian are the same as defined in Equations (12) and (14).

### 4.2.2. QAOAH

For this scenario, it is more challenging to construct the mixing Hamiltonian to describe how to move in the constrained space in QAOA. Unlike the Lagrange multiplier's case, it is not easy to linearly combine two constraint encodings to obtain the final constraint Hamiltonian. In other words, one cannot add $H_{m1} + H_{m2}$ to obtain $H_{m1+m2}$, where $H_{m1}$ $H_{m2}$, $H_{m1andm2}$ are mixing Hamiltonian's representing 1st, 2nd and 1st and 2nd constraints, respectively. This scenario presents two challenges:

1. If a mixing operator allowed a resource in $\mathbf{R}_1$ to move from an unallocated state to an allocated state by pairing it with a mission, then it must also move a resource from $\mathbf{R}_2$ to the same mission.

2. The mixing operator must operate such that it is possible after multiple applications to visit every classical state from the starting quantum state. These two obstacles require a slightly different mixing operator and more qubits.

The strategy for creating a two-constraint mixing operator is to reallocate resources in pairs—one from $\mathbf{R}_1$ and one from $\mathbf{R}_2$. These pairs will always move together. However, the problem is that not every possible classical state can be produced from the mixing. For example, let us say that there are resources $\mathbf{R}_1 = \{1, 2, 3, 4\}$ and $\mathbf{R}_2 = \{5, 6, 7, 8\}$. If resource 1 and resource 5 move together, then it is never possible to see resource 2 just be paired with resource 5 without also being paired with 1. A way to resolve this problem is to introduce an additional mixing operator to swap entire columns within just $\mathbf{R}_1$ or $\mathbf{R}_2$ resources. However, every time columns are swapped, the classical state must remember what mappings are paired with others so that if another reallocation is performed, the buddy constraint will not be violated.

For this reason, new qubits are introduced to each column. These qubits represent the pair ID that is present in the columns for resources in $\mathbf{R}_1$ and $\mathbf{R}_2$. Resources with the same ID are reallocated together. When columns are swapped, the pair IDs of the columns are also swapped.

Consider an MCO with three missions (plus the unallocated mission) and eight resources evenly split between $\mathbf{R}_1$ and $\mathbf{R}_2$. Our initialized state is shown in Figure 5a. Each column has two extra ID qubits (lowest two rows) with a unique bit-encoding that matches another column from the opposite qualification type. This means that these columns are paired together.

For example, consider the case where the mixer operation reallocates resource 2 from the unallocated mission to mission 2. Since this column has an ID of '10$_b$' (top ID qubit true, bottom one false), it is paired with resource 6 because it has the same ID. In order to respect this pairing, both resource 2 and 6 are swapped together using a dual swap gate C-DSWAP, as shown in Figure 5b.

Consider the situation when the mixing operator performs a column swap between columns 2 and 4 (see Figure 5c). Note that the IDs of these columns are also swapped. If the mixer chooses to move resource 6 back to the unallocated state, it also would move resource 4 into the unallocated state since they have the same ID. This can be ensured if the mixer uses the dual swap operation once again (see Figure 5d).

The added qubits to represent the IDs of each of the columns are notated as $|\mathsf{ID}\rangle$ and defined as:

$$|\mathsf{ID}\rangle = \bigotimes_{j_1=0}^{\mathsf{ID}_{max}} |j_1\rangle \otimes \bigotimes_{j_2=0}^{\mathsf{ID}_{max}} |j_2\rangle. \tag{25}$$

The starting state $|\psi\rangle$ is:

$$|\psi\rangle = |\mathsf{ID}\rangle \otimes \bigotimes_{(m,r)\in\mathbf{M}\times\mathbf{R}} \begin{cases} |1\rangle & m = U \\ |0\rangle & m \neq U \end{cases}, \tag{26}$$

and the mixing Hamiltonian $H_m$ becomes

$$H_m = \sum_{p \in \mathbf{R}_1 \times \mathbf{R}_2 \times \mathbf{M}} \sum_{j=0}^{\mathsf{ID}_{max}} \text{C-DSWAP}(p, j)$$

$$+ \frac{1}{2} \sum_{r_1 \in \mathbf{R}_1} \sum_{\substack{r'_1 \in \mathbf{R}_1 \\ r_1 \neq r'_1}} \text{COL-SWAP}(r_1, r'_1) \qquad . \qquad (27)$$

$$+ \frac{1}{2} \sum_{r_2 \in \mathbf{R}_2} \sum_{\substack{r'_2 \in \mathbf{R}_2 \\ r_2 \neq r'_2}} \text{COL-SWAP}(r_2, r'_2)$$

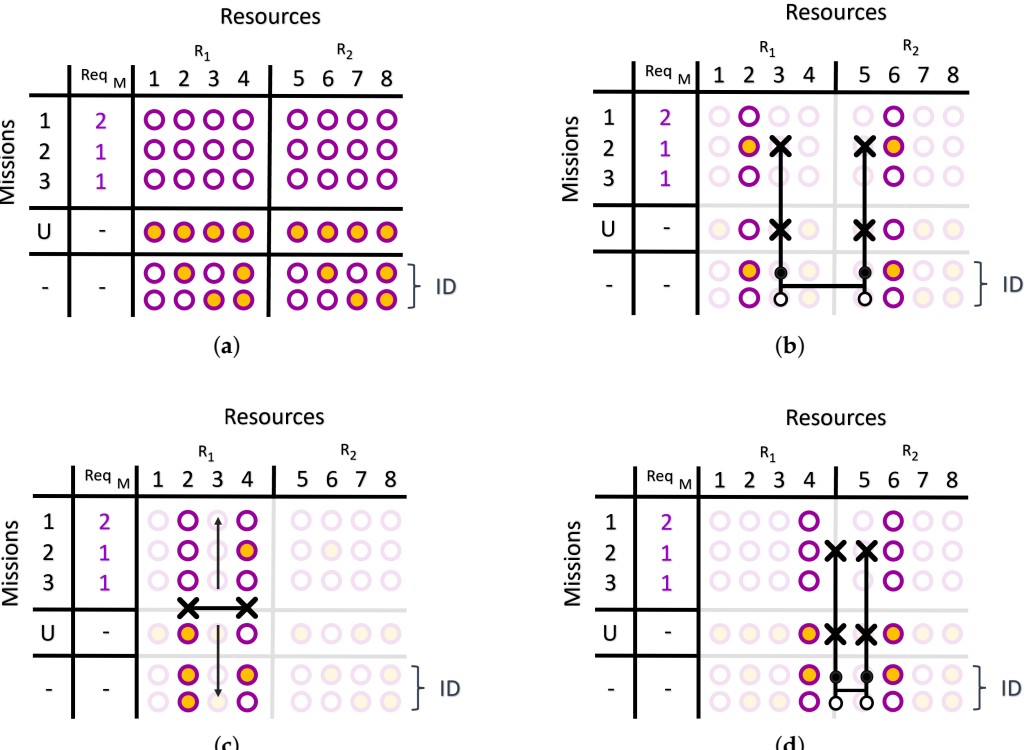

**Figure 5.** (**a**) Initial State Layout. The initial state is similar to the QAOA case except with additional ID qubits to be used to support the constraint mixer. Each column of ID qubits forms a unique binary number for that resource within its resource set. (**b**) First Control Dual-Swap Operation. Through mixing, resource 2 is allocated to mission 2. In order to not violate the row constraint, resource 6 from $\mathbf{R}_2$ is allocated using a dual-swap gate with controls on the ID qubits. (**c**) Column Swap. The mixing operator can never produce a valid solution where just resource 4 and 6 are paired to mission 2 using just Controlled Dual-Swap Operations. Therefore, column-swap operations are permitted, which swap any two columns in the table within its resource set. Here, resource 2 and 4 in resource set $\mathbf{R}_1$ have their entire columns swapped. (**d**) Second Control Dual-Swap Operation. This operation is exactly the same as (**b**), but the Control Dual-Swap operator is used to unallocate resource 4 from mission 2 to mission $U$. Notice that the Control Dual-Swap operator has its control configuration identical on both resource 4 and 6 ID-qubit columns; this is to ensure that the row constraint cannot be violated. A violation occurs when leaving resource 6 with no pair on mission 2.

The constant $\mathsf{ID}_{max}$ represents the maximum required binary states to represent all columns. This is the number of resources in either $\mathbf{R}_1$ or $\mathbf{R}_2$, and it is represented by $\mathsf{ID}_{max}$:

$$\mathsf{ID}_{max} = |\mathbf{R}_1| = |\mathbf{R}_2|. \qquad (28)$$

The column-swap gate, notated as COL-SWAP$(r, r')$, swaps the columns represented by resources $r$ and $r'$. Its decomposition is trivial as it employs many swaps tensored together.

The control dual-swap, notated as C-DSWAP$(p, j)$, has parameters $p$ and $j$. The parameter $p = (r_1, r_2, m)$ is a tuple composed of a resource from $\mathbf{R}_1$, a resource from $\mathbf{R}_2$, and a mission $m$ in $\mathbf{M}$. This gate applies a control swap gate to resource $r_1$ and $r_2$ between the mission $m$ and the unallocated mission U. The parameter $j$ is an ID which represents how to control the swap gate. For example, $10_b$ is applying a control-true, control-false gate to both IDs in the columns represented by $r_1$ and $r_2$.

To implement the algorithm on the IBM machine, we must decompose $H_m$ such that it is a sum of tensored Pauli gates. The column-swap gates COL-SWAP$(r, r')$ can be decomposed, knowing that they are made up of swap gates, as from Equation (18).

Decomposing a generalized version of the control dual-swap gate is tedious, so we provided an example decomposition for C-DSWAP$(*, 10_b)$, which is the gate used in Figure 5b,d. First, we present the Pauli-decomposition of the control-true and control-false unitary operations shown in Equations (29) and (30), respectively. Unitary $A$ is arbitrarily acting on $m$ qubits.

$$\text{C-T}(A) = \frac{1}{2}\big((I + Z) \otimes I^{\otimes m} + (I - Z) \otimes A\big). \tag{29}$$

$$\text{C-F}(A) = \frac{1}{2}\big((I - Z) \otimes I^{\otimes m} + (I + Z) \otimes A\big). \tag{30}$$

Now, C-DSWAP$(*, 10_b)$ can be represented in terms of control-true and control-false unitaries and the dual-swap gate DSWAP:

$$\begin{aligned} \text{C-DSWAP}(*, 10_b) = \\ \text{C-T}(\text{C-F}(\text{C-T}(\text{C-F}(\text{DSWAP})))). \end{aligned} \tag{31}$$

Following this, the dual-swap gate DSWAP is two swap gates tensored together:

$$\text{DSWAP} = \text{SWAP} \otimes \text{SWAP}. \tag{32}$$

The DSWAP can further be decomposed using Equation (18).

As mentioned in the previous section, our two-constraint MCO problem has permutation symmetry between the resources in the same set/group. Therefore, the column swap terms can be effectively removed, and $H_m$ becomes

$$H_m = \sum_{p \in \mathbf{R}_1 \times \mathbf{R}_2 \times \mathbf{M}} \sum_{j=0}^{\text{ID}_{max}} \text{C-DSWAP}(p, j). \tag{33}$$

This effectively shrinks the search space from the total constraint space. However, because of the symmetry, it is known that the optimal solution still lies inside the smaller subspace. When different capabilities are introduced to each resource, this optimization technique cannot be used since the permutation symmetry is not guaranteed.

## 5. Analyses of Results

In this section, we compare the results of the different MCO implementations. Employing the implementation methods discussed above, the MCO problem was run on the D-Wave and on IBM machines, capturing several key metrics:

- Number of qubits,
- Quantum processor time,
- Cost,
- Number of constraints violated.

For both scenarios (see Table 1), 100 random MCO configurations were generated using up to 27-qubits (200 different MCO configurations in total). Quantum Annealing, QAOA, QAOAH, and Brute Force (BF) methods were run for each generated configuration. For Quantum Annealing, the DW_2000Q_6 machine was used, while ibmq_toronto, ibm_hanoi, ibm_cairo, ibmq_mumbai, and ibmq_montreal machines were used for running QAOA and QAOAH. The Lagrange multipliers were set to five for both scenarios, and a *p*-value of two is used for QAOA (this parameter is discussed in the original paper [17]). The Quantum Annealing runs each sample of the anneal 50 times, while each IBM job sampled the state-vector 1000 times. These parameters were chosen based on a good balance of timing, cost, and constraints satisfied found by preliminary results not discussed in this paper.

**Table 1.** S1 and S2 stands for Scenario 1 and 2, respectively. MCO implementations in hardware and simulation execution environment. And number of constraints violated in the various cases.

| Results Machine | Execution Environment | | | Average Violated Constraints | | |
|---|---|---|---|---|---|---|
| | **D-Wave** | **IBMQ** | | **D-Wave** | **IBMQ** | |
| **Algorithm** | **QA** | **QAOA** | **QAOAH** | **QA** | **QAOA** | **QAOAH** |
| S1 Hardware | ✓ | ✓ | ✓ | 0.05 | 0.97 | 3.89 |
| S1 Simulation | - | - | - | - | - | - |
| S2 Hardware | ✓ | ✓ | - | 0.0 | 0.96 | - |
| S2 Simulation | - | - | ✓ | - | - | 0.0 |

*5.1. Scenario 1*

Figure 6 shows the timing averages for QA, QAOA, and QAOAH, respectively, versus the problem qubit size.

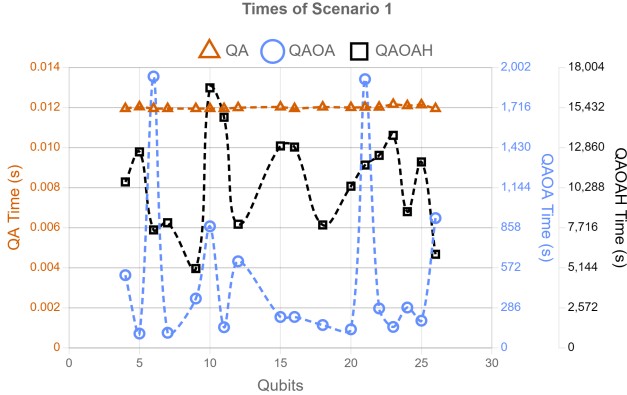

**Figure 6.** QPU Results QPU Times for Scenario 1 Runs. QA timing includes the anneal time of each 50 samples, 20 microseconds per sample. QAOA and QAOAH varied in the number of jobs that ran, each of which ran the parameterized circuit 1000 times.

Because the timing differed by different levels of magnitude, three y-axes with different scales are displayed. QA run time outperforms the other methods by having an overall constant run-time of 0.012 s, while QAOAH can use the QPU for 5 h across all jobs in certain worse-case instances (this excludes queue-time and creation time on QPU for IBMQ devices). For all methods, timing was calculated based on qpu time (not wall-clock time). For the QA method, 'qpu_sampling_time' was used to calculate the total qpu time, while 'running time' is used for QAOA and QAOAH.

Figure 7a shows the average costs for all MCO configurations versus the number of qubits it took to encode each scenario 1 run. These costs do not include the cost accumulated from the embedded constraint functions Equations (9) and (24) as these are used to make the constraints hold. Furthermore, all the costs plotted are actually the cost for that particular run minus the best possible cost it can receive. This best possible cost is computed by using

brute force search methods in simulation. This difference is referred to as the relative cost. Therefore, the best possible relative cost a run could have is zero. For most runs, it can be seen that QA and QAOA runs have a relative cost of around zero, but as the number of qubits increase, QAOA becomes less optimal compared to QA. The QAOAH approach, when using a large amount of qubits, actually had a negative relative cost indicating that it must have violated some constraints in order to achieve a cost below the solution found with brute-force.

In Figure 7b, the average number of constraints violated is plotted against qubit size. The number of constraints is calculated by counting the number of resources that were assigned to more than one mission. For example, if resource 1 was assigned to two extra missions, and resource 2 was assigned to three extra missions, then the number of constraints violated is five. QA mostly had no constraints violated at any sized qubit problem, while QAOA and QAOAH suffered constraint violations when the problem size increased.

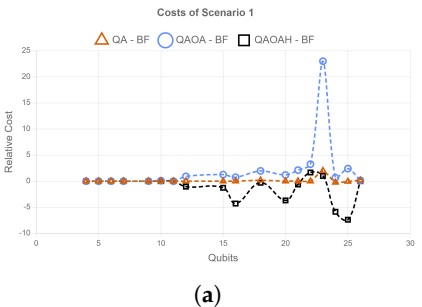
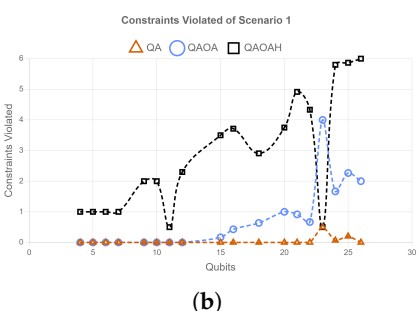

(**a**)  (**b**)

**Figure 7.** QPU Results (**a**) Average relative cost versus the number of qubits used. The relative cost is the mission plus the precedence cost obtained minus the lowest possible mission plus precedence cost that can be achieved (while keeping constraints) for the problem; the lowest possible cost is found by brute force: Relative cost = algorithm's solution cost–brute force solution's cost. Negative relative costs imply that the solution violated constraints. (**b**) Average number of constraints violated versus the number of qubits used. The number of constraints counts up how many extras qubit are on within a column in the matrix view representation. This count includes the number of columns that do not have any qubits on it at all.

*5.2. Scenario 2*

For scenario 2, the QAOAH method was calculated via IBM's state-vector numerical simulation tool due to the long running-time of runs in scenario 1. Because of this shift from running on actual hardware (scenario 1), to simulation (scenario 2), the QAOAH method is plotted using its wall-clock times against QA's and QAOA's QPU time in Figure 8. Even with this change, the magnitudes of running-time are very diverse, so a third y-axis is added, as before. Furthermore, for the QAOAH approach, the mixing operator in Equation (33) is used instead of Equation (27) because removing the column-swap terms reduces the total gate count of the overall computation, making simulation times faster.

In scenario 2, QA times are quite faster than QAOA methods. It can be seen that the QAOAH method has far less flexibility in terms of qubit-range. This is because in our MCO algorithm implementation, extra qubits are required to represent each resource's ID. To keep consistency with the QA and QAOA methods, the x-axis in each plot in this section represents the number of qubits used minus the amount used to represent the IDs.

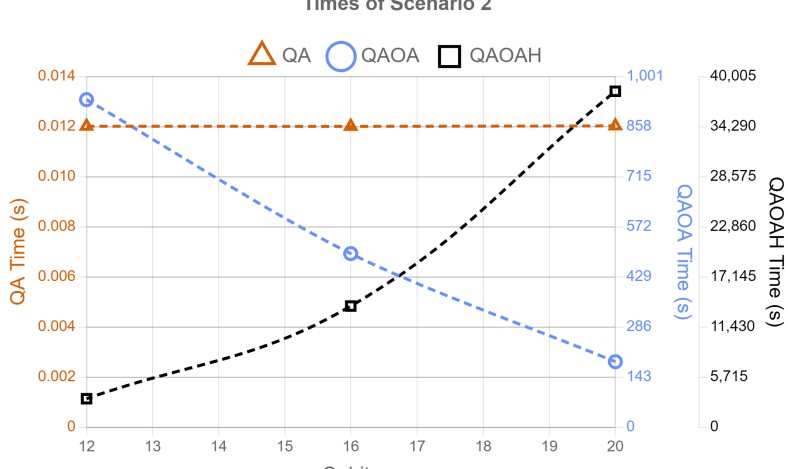

**Figure 8.** QPU and Simulated Results QA and QAOA QPU times and simulated QAOAH wall-clock times for Scenario 2. QA timing includes the anneal time of each 50 samples, each 20 microseconds per sample. QAOA ran with 1000 shots. QAOAH ran an undetermined amount of jobs using the statevector-simulator to obtain results. Unlike Scenario 1, the times recorded are total wall-clock times for QAOAH.

As in scenario 1, the relative costs for each method are plotted in Figure 9a. Both QAOAH and QA methods have zero relative costs. However, QAOA by itself did not exhibit a positive relative cost. For both cases in this subplot, the data are insufficient to deduce whether or not these violated constraints.

In Figure 9b, the average number of constraints violated is plotted against the number of qubits the problem encodes. The number of constraints violated is calculated similarly to scenario 1, but now including all violations of the second constraint. For example, if four resources of type-1 and 2 of type-2 were allocated to a mission, then the number of constraints violated is $4 - 2 = 2$. Figure 9b shows that QA and QAOAH did not violate constraints at any qubit size, while QAOA did on average. QAOA violated constraints mostly likely because it found a solution where the mission cost exceeded the cost incurred by the constraint function. QAOAH in simulation, however, did not violate constraints because the mixing operator that was used transforms solutions without leaving the constrained space. These results contrast with the QPU runs for QAOAH in Scenario 1 where it did violate constraints. Since the simulation ran without any noise profiles, it is expected that QAOAH should not violate constraints in theory, but this is not the case when running on the actual quantum machine. The source of noise on actual hardware is most likely due to the gate noise and noise from measurement.

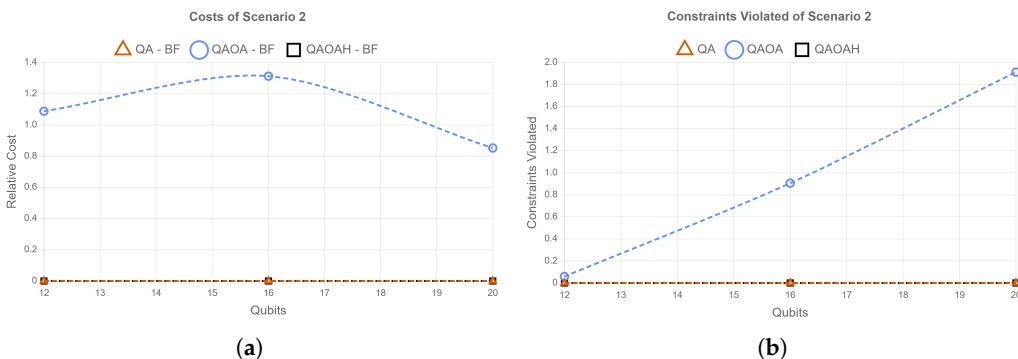

**Figure 9.** QPU and Simulated Results (**a**) Average relative cost versus the number of qubits used. The relative cost is the mission cost obtained minus the lowest possible mission cost that can be achieved (while keeping constraints) for the problem. The lowest possible cost is found by brute force. Negative relative costs must mean that the solution violated constraints. (**b**) Average number of constraints of each method versus the number of qubits used. The number of constraints counts up how many extra qubits are within a column in the matrix view representation plus the absolute difference of qubits within a row between resource sets ($\mathbf{R}_1$ and $\mathbf{R}_2$). This count includes the number of columns that do not have any qubits on at all.

## 6. Summary and Conclusions

In this paper, we introduced Mission Covering Optimization (MCO), implemented three different constrained optimization techniques (QA, QAOA, and QAOAH) to find solutions to two scenarios of MCO, and discussed results after running implementations on the IBM machine, D-Wave Machine, and on a state vector simulator. Results were compared based on timing, relative cost, qubits used, and constraints violated. From the 200 tests performed on each scenario, QA achieved the quickest results while using the least number of qubits and violating the least number of constraints. We conjecture that QAOA and QAOAH approaches may have taken longer for the gradient descent algorithm to be convinced that an optimal solution was found because of the abundance of noise in current hardware. It was found that it is nontrivial to engineer multiple constraints by embedding in the mixing-Hamiltonian, especially when compared to the ease of using Lagrange multipliers, in simulation, where adding constraints entails simply adding terms together. The study conducted here suggests that the additional complexity in the QAOAH approach poses potential scalability challenges (due to the additional qubits required to ID the constraints) for problems similar to MCO with multiple types of unique constraints.

The results of the work presented here illustrate the fact that at the current nascent stage of quantum computing, one is still coding to the quantum computing paradigm; here, Adiabatic Quantum Computing Model (AQCM) (as instantiated by the D-Wave machine), and the Gate-Based Model (GBM) (as instantiated by the IBMQ). Even at the ideal simulation stage of this study, there are substantial differences in the resources required for implementing constraints under the AQCM and the GBM, with the latter requiring many more ancilla qubits. In the AQCM, mixing occurs naturally through the adiabatic evolution, while in the GBM, mixing has to be explicitly implemented, which for the model we investigated, entailed the construction of mixers that would not violate the constraints. In simulation, this construction worked seamlessly; however, in practice, it was limited by the actual noise in the qubit-based implemented quantum circuits involved. This is a well-recognized limitation of today's current noisy intermediate scale quantum (NISQ) hardware.

Lastly, while not directly explored in this study, the network connectivity plays an important role in contributing to the noise (resulting in constraint violation) under each paradigm. It is well-known that under the AQCM, the implementation of constraints requires the 'chaining' together of many individual qubits in order to act as an effective

single qubit. In practice, the length of chain is finite (typically 10–20 qubits) and 'breakage' of the chain, due to local environmental decoherence, acts as another important source of circuit noise and degradation. In the GBM, extra ancilla are required to implement constraints, increasing the circuit depth of the specific problem under study, which is then subject to local environmental decoherence effects.

The goal of the work presented here was to illustrate, albeit in a very specific, small qubit model, the challenges of translating the implementation of a constrained optimization quantum (MCO) algorithm between two quantum computing paradigms, AQCM and GBM, to identify (for this illustrative problem) where the challenges lay for the purpose of tackling larger-scale problems.

## 7. Future Work

Adding further capabilities to different resources in Scenario 2 would make for a more interesting/realistic optimization problems for quantum computers to solve. This is just one of the many alterations that can be applied to MCO to increase the complexity of the optimization problem.

In this paper, resource dependencies are modeled within MCO; however, there may be missions with different priorities along with mission dependencies. Furthermore, in this work, resources were only shown to possess one type of qualification. However, there can be cases where a resource may have many types of qualifications. At the heart of MCO, it is an optimization problem concerning the allocation of resources invariant to time. An interesting direction for future study is how well this type of optimization problem can be ported to an extension of a job-shop problem.

Although error mitigation is not a focus of this study, research for mitigating errors for QAOA is being conducted by others [22]. Employing error mitigation techniques for MCO is a direction of study.

There are alternate techniques that can be employed on gate-based machines to solve QUBO problems. For example, the Grover adaptive search [23] is an iterative Grover-like approach that filters states based on a conditional oracle. Furthermore, filtering-VQE [24] can be used to solve optimization problems, utilizing filtering operators to achieve faster and more reliable convergence to the optimal solution.

Lastly, another interesting solution method might entail the use of a 'bang-bang' strategy [25] for multiple constraints. Here, for each constraint $C_i$ associated with a constraint Hamiltonian $H_i$, one might randomly cycle through applications of individual $H_i$ for each time step, as opposed to the application of the joint Hamiltonian $H = \sum_i H_i$. While each $H_i$ only preserves constraints $C_i$, the supposition is that the application of $H_i$ might only partially violate constraints $C_{j \neq i}$, if applied for a short time and randomly. This solution approach to MCO-like problems will be investigated in future work.

**Author Contributions:** Conceptualization and methodology, all authors contributed; software, M.C., A.G. and A.S.; resources, supervision and funding A.G., P.M.A. and L.W.; writing—review and editing, M.C., A.G. and P.M.A. All authors have read and agreed to the published version of the manuscript.

**Funding:** This research received no external funding.

**Data Availability Statement:** For any request related to the data please contact the corresponding author.

**Acknowledgments:** The authors would like to thank David Vernooy and the exponential campaign at GE Research for supporting this effort, and Daniel Koch and Saahil Patel for useful discussions. The views expressed are those of the authors and do not reflect the official guidance or position of the United States Government, the Department of Defense, the United States Air Force or General Electric. The appearance of external hyperlinks does not constitute endorsement by the United States Department of Defense or General Electric of the linked websites, or the information, products, or services contained therein. The Department of Defense and General Electric do not exercise any editorial, security, or other control over the information you may find at these locations.

**Conflicts of Interest:** The authors declare no conflict of interest.

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
