# Peer review of "Quantum Computing Approaches for Mission Covering Optimization"

_algorithms, doi:10.3390/a15070224_

Round 1

Reviewer 1 Report

The paper defines a combinatorial optimization problem called MCO as a variant of the resource assignment problem and designs algorithms for three types of quantum computers.

I believe it is an interesting report to know the optimization performance of the available quantum annealing machine and NISQ device.

However, I cannot accept the current paper since it does not include enough novelty and valuable results.

To denote the novelty and usefulness of the problem definition of MCO, the author needs to explain the details, for example, the complexity of MCO, the problem is NP-hard?, and why MCO is used for the target problem?

I cannot understand well the objective of the experimental evaluation.

This paper shows some experimental results performed on D-Wave and an IBM quantum computer. However, the current capability of both platforms is different so much. 

Reviewer 2 Report

Review of: “Quantum Computing Approaches for Mission Covering Optimization” by Cutugno, Giani, Alsing, Wessing and Schnore

In this manuscript, the authors studied Mission Covering Optimization (MCO) by implementing Quantum Annealing (QA), Quantum Alternating Operator Ansatz (QAOA) and Quantum Alternating Operator Ansatz with Constraint Hamiltonian (QAOAH). 

This manuscript carried out interesting analysis. 

Let go step by step: 

1. Page 1, Line 36: QPU. What is it? 

2. Page 2, Line 76: “Q = {q_1, ..., q_{N_Q}} is the number of N_Q qualifications in the MCO problem.” --> “N_Q is the number of the qualifications with the set Q = {q_1, ..., q_{N_Q}}.” 

3. Page 2, after Line 78: “are integer number” --> “are integer numbers”

4. Page 3, Line 83: “mission 1 requires 10 resources that can fly a plane” --> “mission 1 requires n resources that can fly a plane (q_1)”

5. Page 3, Line 91: “on scenario 2.” What is scenario 2? Not yet clear until this spot in the manuscript.

6. Page 3, Line 99 and after: “These scenarios are not computationally difficult to compute classically. However, these scenario’s design is intentional for comparing results in this study, ...” --> Classically, not difficult. Then, what is the benefit if considering its quantal counterpart discussed in this manuscript? I do not find, from this manuscript, any clear-cut superiority of this quantum approach to its classical counterpart. Please clarify it.

7. Page 3, Line 100: “these scenario's design is ...” --> “the design of these scenarios is ...”

8. Page 4, Line 102: “can be used to without ...” --> “can be used without ...” AND “since the scenario's are ...” --> “since the scenarios are ...”

9. Page 4, Line 116: “is composed by” --> “is composed of”

10. Page 20, Line 604: This line should be removed.

11. This manuscript provides the “straightforward” results of quantal MCO. But I would like to ask, “So what?” Please add some words, if any, about “scientific” meaning of their findings (for example, into “Summary ad Conclusion”).

In summary, let me receive the revised version of the manuscript meeting the above-mentioned conditions and then decide the qualification of this manuscript for publication.

Reviewer 3 Report

Cutugno et al. apply quantum optimization algorithms based on QUBO and QAOA formulations to tackle a variant of constraint resource allocation problem.

The paper is relevant to the journal's scope and its results are novel and interesting, at least to my eyes. The paper is overall well-structured and the presentation of the results is very good.

In my opinion, a big weakness of the submitted paper is its abstract. The starting expression "We study quantum computing algorithms" is too general and actually doesn't correspond to what the paper discusses. Also, it's not a good practice to use references there, I suggest removing them. Finally, a comment on the achieved results is missing. That is, what do the results of the different algorithms showed? What about the analysis the authors performed? These things are necessary in the abstract and will only enhance the paper's strengths and outcomes.

Then, section 1 is also incomplete, which in turn weakens the transition to section 2. Some key concepts introduced in the first section should be explained a bit more.  Adding references there, with a brief description of what QUBO and QAOA approaches are (and where they are usually used and applied, eg. for what kind of problems they are applied and are good at), would benefit the paper and help transitioning to next parts. Otherwise, one has to reach section 4, but this is too late in my opinion.

A big issue with the manuscript is the graphics quality. For example, the colour choice in the plots of Figs 6-8 is bad and makes it difficult for the readers. Especially if visual impairment (such a colour blindness) is taken into consideration. But I have to say that their caption is appropriate and as a whole, those plots are very good at delivering the message. 

Table 1 should also be reformatted. The gray background doesn't help at all.

Some typos and styling remarks:

The paper has a lot of typos and one can spot a careless writing. I would recommend a good and detailed proofreading during the revision stage.

line 152 it’s 

line 20 space before reference [2]. This typo is actually found across the text, so i highly recommend a thorough proofreading before resubmission.

line 35 computer -> computers

there is a second list of references on the last page, probably misplaced from the template file.

Reviewer 4 Report

The authors study quantum computing algorithms for solving constrained resource allocation

problems and compare solutions to optimization problems using three different procedures. Results were compared based on timing, relative cost, qubits used, and constraints violated. The authors present numerical results of 200 tests performed on each scenario, They also propose future research to improve the optimization techniques.

The manuscript is very well organized and provides a detailed description of the optimization procedures. I can only suggest minor revisions to improve it. When introducing mission and resource requirements on page 3 the following words “and represents the qualification requirement for each mission” are repeated twice (which is probably incorrect?). When the authors start to discuss an example in Fig. 1 on page 4 they should probably add more details about the example – the “pilots” appear out of nowhere. A few words of explanation in the definition of the capability function on the same page could be useful.

To summarize, I recommend this manuscript for publication after minor revision.

Round 2

Reviewer 1 Report

The improved manuscript includes valuable information for researchers of quantum informatics. 

Reviewer 2 Report

This manuscript is ready for publication now.

Reviewer 3 Report

First of all I thank the authors for taking the time to respond to my comments and I hope they found them useful and constructive. In my opinion the authors revised their manuscript sufficiently (eg the new abstract is quite improved now and I feel it gets the message properly, introduction is smoother, and so on) and have addresses all of my questions and criticism. 

In my opinion, the paper can now be published.